# Adapting COVID-19 research infrastructure to capture influenza and respiratory syncytial virus alongside SARS-CoV-2 in UK healthcare workers winter 2022/23: Results of a pilot study in the SIREN cohort

Sarah Foulkes[1][☯]*, Katie Munro[1][☯], Dominic Sparkes[1], Jonathan Broad[1], Naomi Platt[1], Anna Howells[1], Omolola Akinbami[1], Jameel Khawam[1], Palak Joshi[1], Sophie Russell[1], Chris Norman[2], Lesley Price[3], Diane Corrigan[4], Michelle Cole[1], Jean Timeyin[1], Louise Forster[1], Katrina Slater[1], Conall H. Watson[1], Nick Andrews[1], Andre Charlett[1], Ana Atti[1], Jasmin Islam[1], Colin S. Brown[1], Jonathan Turner[1], Susan Hopkins[1], Victoria Hall[1], SIREN study group[¶]

1 United Kingdom Health Security Agency (UKHSA), London, United Kingdom, 2 Health and Care Research Wales, Cardiff, Wales, 3 Public Health Scotland, Glasgow, United Kingdom, 4 Health and Social Care in Northern Ireland (HSC), Belfast, United Kingdom

¶ Membership of the SIREN Study Group is listed in the Supplementary Material.
☯ These authors contributed equally to this work.
* Sarah.Foulkes@UKHSA.gov.uk

## Abstract

### Introduction

The combination of patient illness and staff absence driven by seasonal viruses culminates in annual "winter pressures" on UK healthcare systems and has been exacerbated by COVID-19. In winter 2022/23 we introduce multiplex testing aiming to determine the incidence of SARS-CoV-2, influenza and respiratory syncytial virus (RSV) in our cohort of UK healthcare workers (HCWs).

### Methods

The pilot study was conducted from 28/11/2022–31/03/2023 within the SIREN prospective cohort study. Participants completed fortnightly questionnaires, capturing symptoms and sick leave, and multiplex PCR testing for SARS-CoV-2, influenza and RSV, regardless of symptoms. PCR-positivity rates by virus were calculated over time, and viruses were compared by symptoms and severity. Self-reported symptoms and associated sick leave were described. Sick leave rates were compared by vaccination status and demographics.

**Data availability statement:** Anonymised data will be made available for secondary analysis to trusted researchers upon reasonable request.

**Funding:** The SIREN study was funded by the UK Health Security Agency; the UK Department of Health and Social Care with contributions from the governments in Northern Ireland, Wales, and Scotland; the National Institute for Health Research; Health Data Research UK (NIHR200927; HDRUK2022.0322). The funders had no role in study design, data collection and analysis, decision to publish, or preparation of the manuscript.

**Competing interests:** NO authors have competing interests.

## Results

5,863 participants were included, 84.6% female, 70.3% ≥45-years, 91.4% of White ethnicity and 82.6% in a patient facing role. PCR-positivity peaked in early December for all three viruses (4.6 positives per 100 tests (95%CI 3.5, 5.7) SARS-CoV-2, 3.9 (95%CI 2.2, 5.6) influenza, 1.4 (95%CI 0.4, 2.4) RSV), declining to <0.3/100 tests after January for influenza/RSV, and around 2.5/100 tests for SARS-CoV-2. Over one-third of all infections were asymptomatic, and symptoms were similar for all viruses. 1,368 (23.3%) participants reported taking sick leave, median 4 days (range 1–59). Rates of sick leave were higher in participants with co-morbidities, working in clinical settings, and who had not been vaccinated (COVID-19 booster or seasonal influenza vaccine) versus those who had received neither vaccine (2.04 vs 1.41 sick days/100 days, adjusted Incidence Rate Ratio 1.47 (95%CI 1.38, 1.56).

## Conclusion

This pilot demonstrated the use of multiplex testing allowed better understanding of the impact of seasonal respiratory viruses and respective vaccines on the HCW workforce. This highlights the important information on asymptomatic infection and persisting levels of SARS-CoV-2 infection.

## Introduction

UK healthcare systems experience significant challenges each winter, due to the impact of seasonal surges in respiratory viruses, including influenza, respiratory syncytial virus (RSV) and since 2020, COVID-19, which combined are often described as "winter pressures" [1,2].

While all three viruses contribute to winter pressures, the epidemiology and vaccination approach for each differs. Both influenza and RSV are most common during the winter period, while COVID-19 does not exhibit a predictable seasonality [3–5]. Seasonal influenza vaccination is recommended for frontline healthcare workers (HCWs) each winter [6]. Since December 2020, vaccinations for COVID-19 have been available and prioritised for HCWs, and in winter 2022, HCWs were offered a booster dose [7]. In winter 2022/23, RSV vaccination was not available for HCWs [5].

Respiratory illness increases patient attendance but also causes significant staff absence, with respiratory illness the second most common cause of sick leave within the NHS [8]. Reduction in staff sickness absence and preventing nosocomial transmission are the rationale for the annual NHS winter flu vaccine campaign for HCWs and the prioritisation of HCW for COVID-19 vaccine boosters [3,9,10].

As a result of the pandemic, COVID-19 has been well characterised in relation to key scientific and clinical questions including the role of antibodies as correlates of protection, rates of asymptomatic disease and real-world vaccine effectiveness [11–16]. However, this is not the case for the other principle seasonal viruses, influenza and RSV, where the focus has been on symptomatic disease and in populations

most at risk, including children and the elderly [17,18]. HCWs are an important population to study given the impact of respiratory illnesses on the workforce and their high exposure [19,20], offering insights into the natural history, epidemiology and burden of influenza and RSV in working aged populations.

The SARS-CoV-2 Immunity and Reinfection Evaluation (SIREN) study is a prospective cohort study of HCWs across the UK, with participants completing regular SARS-CoV-2 PCR testing and antibody testing continuously since June 2020 [21]. It was set up at the start of the pandemic to assess the risk of re-infection with SARS-CoV-2 and has continued to adapt to address key scientific questions.

During the COVID-19 pandemic, the introduction of non-pharmaceutical measures, including universal masking and social distancing, was found to reduce the rates of other respiratory viruses [3,22]. However, following the removal of these interventions, the winter of 2022/23 was the first opportunity to understand the impact of circulating seasonal viruses on the NHS workforce. Therefore, in winter 2022/23 the SIREN study piloted a "Winter Pressures sub-study" that included multiplex PCR testing for influenza and RSV alongside SARS-CoV-2 [23].

This paper outlines the results of the SIREN Winter Pressures Pilot study 2022/23, with the aim of a) describing the burden of influenza (A and B), RSV, and SARS-CoV-2, b) characterisation of SIREN HCW symptom profile, and c) understanding time off work due to being symptomatic in the context of HCW vaccination.

## Methods

### Study design

The SIREN Winter Pressures sub-study is a prospective cohort study nested within the SIREN UK multicentre HCW cohort study [21,23].

### Participants

Participants were recruited into the sub-study via two routes: 1) participants previously completing monoplex testing (for SARS-CoV-2 only) were informed, in writing, before the sub-study start date (28 November 2022) that testing would move to multiplex testing (SARS-CoV-2, Influenza and RSV), and were given the opportunity to withdraw from the study; 2) additional participants were re-recruited (from participants who were not currently testing) and consented directly into the winter pressures postal testing pathway, between 13 December 2022 and 24 January 2023 [18].

Participants undergoing PCR testing between 28 November 2022 and 31 March 2023 were included in the sub-study. Participants contributed different lengths of time to the analysis period, due to completing study follow-up time or withdrawing from the study.

### Data collection

Participants completed fortnightly questionnaires on symptoms (onset date, type and duration) and related time taken-off work due to reported symptoms, in addition to an enrolment survey at the start of the SIREN study, detailing the participant's demographics, occupation, underlying medical conditions and household composition. Vaccination data (COVID-19 and seasonal influenza vaccination) was obtained both from the fortnightly questionnaire and linkage to national vaccination registries.

Participants completed swabs for multiplex PCR (SARS-CoV-2, influenza and RSV), fortnightly regardless of symptoms. PCR testing, symptoms and sociodemographic data were linked via a unique study ID.

### Variables

We defined participants who reported at least one of any of the following symptoms as being symptomatic: cough, fever, shortness of breath, sore throat, runny nose, headache, muscle aches, altered sense of smell or taste, fatigue, diarrhoea, nausea or vomiting, itchy red patches on fingers or toes, rash, swollen glands [23]. Influenza-like illness (ILI) was defined

as participants reporting fever and at least one of following: cough, sore throat, shortness of breath, headache, muscle ache or fatigue. For this analysis, COVID-19 vaccination refers to receiving the second booster dose of the COVID-19 vaccine, and influenza vaccination refers to receiving the 2022/23 seasonal influenza vaccine between 01 September 2022 and 31 March 2023. Sick leave was defined as any self-reported days off work due to being symptomatic.

## Outcomes

The co-primary outcomes were a) a PCR-positive test for either SARS-CoV-2, influenza or RSV; b) proportion of participants reporting ILI symptoms; and c) the number of days taken off work due to being symptomatic.

## Inclusion criteria

Participants who completed at least one fortnightly questionnaire and at least one PCR test during the analysis period were included.

## Statistical analysis

Participants' sociodemographic and occupational characteristics were described.

Infection rates by infection were calculated over time. PCR positive samples were de-duplicated resulting in one sample per fortnight per participant, prioritising a positive sample over a negative sample in the same fortnight. Samples which were positive for multiple viruses were excluded from the analysis to reduce bias in comparing trends and symptom profiles across infections. Further de-duplication by infection episode meant that for SARS-CoV-2, participants could only have one positive sample per 90 days. For influenza and RSV, participants could only have one positive sample per 30 days.

For the infection analysis, participants were considered vaccinated if they had received the seasonal vaccine dose at least 14 days before the first positive PCR sample of the infection episode.

An infection was considered symptomatic if the participant had reported a symptom onset date seven days pre or post the date of first positive PCR sample for each infection episode.

The proportion of participants with each infection was summarised by sociodemographic characteristics and vaccination status. Symptom type, number, duration, hospital attendance and time off work were compared by infection and vaccination status (SARS-CoV-2 and influenza only) using proportions.

The proportion of participants reporting ILI symptoms was calculated over time, by dividing the number of surveys where ILI was reported by the total number of surveys in each time period.

Sick leave rate was calculated by fortnight, using the number of days taken off work divided by the number of days participants contributed to the analysis period, per 100 days, with 95% confidence interval. Incidence rate ratio were calculated to estimate demographic factors associated with taking time off work, unadjusted and adjusted (vaccination status, age, occupational setting and co-morbidities).

For sick leave rate by vaccination status, participants contributed days to the vaccinated state (COVID-19 and influenza separately or simultaneously) if they had received the vaccine at least 14 days before the start of the fortnightly survey period. Participants contributed days to the unvaccinated state if they had not received the seasonal dose or had received the dose after the end of the survey period. For participants who received either vaccine within a survey period, or within the 14 days prior to the survey start date, this survey period did not count towards the rate analysis.

## Ethics statement

The SIREN study was approved by the Berkshire Research Ethics Committee (IRAS ID 284460, REC Reference 20SC0230) on 22 May 2020. The Winter Pressures sub-study was supported by two ethics amendments on 14 November 2022 and 01 December 2022. Participants were informed in advance, as the frequency and method of sampling remained the same, implied consent processes were approved by the committee. Participants returning to the study give informed consent. Clinical trial registration number: ISRCTN11041050; registration date: 12 January 2021.

## Results

A total of 7,774 participants consented to the SIREN Winter Pressures pilot sub-study from 28 November 2022 to 31 March 2023. Of these, 5,863 (75.4%) participants had both survey and testing data and were included in the analysis (Fig 1). Participants were excluded if they withdrawal and requested their data to be removed from the study (n = 18), if they did not complete a follow-up survey (n = 917) or PCR sample (n = 1,789) within the study period. Most participants were female (84.6%), of white ethnicity (91.4%), over 45-years of age (70.3%) and 26.4% reported having a comorbidity. The largest staffing group were nurses (33.4%) followed by administrative (17.3%) and doctors (11.4%). Non all participants were employed in clinical roles, however 82.6% of participants stated they were in a patient facing role. Study participants were highly vaccinated, with 74.2% receiving both COVID-19 and the seasonal influenza vaccine (Table 1).

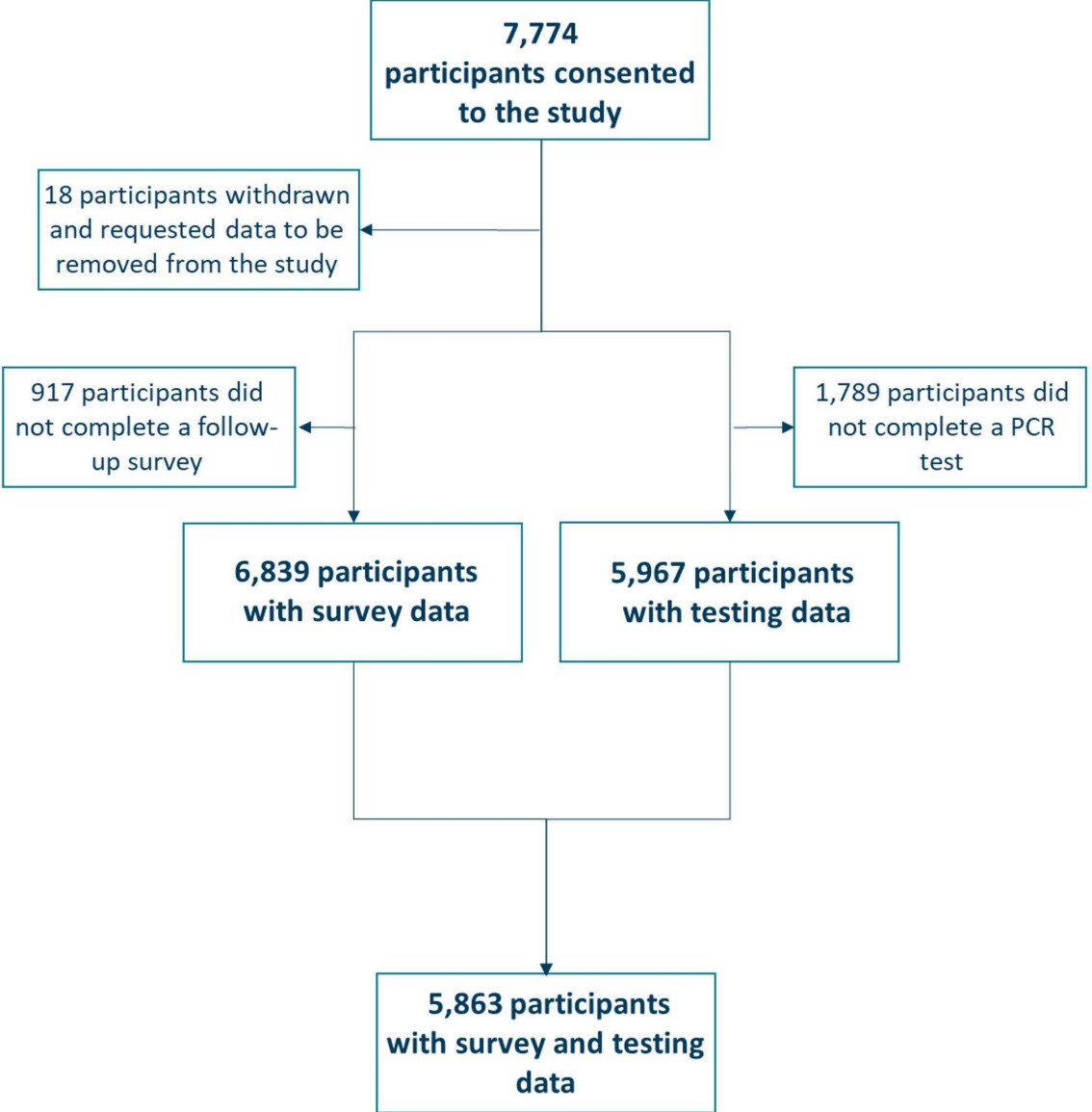

**Fig 1. Flow diagram describing participants included in the SIREN Winter Pressures pilot sub-study.**

**Table 1. Demographics of participants included in the analysis, overall and by infection (SARS-CoV-2, influenza and RSV), 28 November 2022 to 31 March 2023.**

| Characteristic | All participants n (%) | SARS-CoV-2 positive n (%) | Influenza positive n (%) | RSV positive n (%) |
|---|---|---|---|---|
| **Total** | **5,863** | **643** | **91** | **74** |
| Age group | | | | |
| Under 25 | 39 (0.7) | 1 (0.2) | 1 (1.1) | 3 (4.1) |
| 25–34 | 447 (7.6) | 56 (8.7) | 11 (12.1) | 2 (2.7) |
| 35–44 | 1,257 (21.4) | 151 (23.5) | 24 (26.4) | 13 (17.6) |
| 45–54 | 2,280 (38.9) | 244 (37.9) | 30 (33.0) | 36 (48.6) |
| 55–64 | 1,690 (28.8) | 173 (26.9) | 23 (25.3) | 20 (27.0) |
| Over 65 | 150 (2.6) | 18 (2.8) | 2 (2.2) | 0 (0.0) |
| Gender | | | | |
| Female | 4,960 (84.6) | 541 (84.1) | 67 (73.6) | 64 (86.5) |
| Male | 896 (15.3) | 101 (15.7) | 24 (26.4) | 9 (12.2) |
| Non-binary | 4 (0.1) | 1 (0.2) | 0 (0.0) | 1 (1.4) |
| Prefer not to say | 3 (0.1) | 0 (0.0) | 0 (0.0) | 0 (0.0) |
| Ethnicity | | | | |
| White | 5,359 (91.4) | 594 (92.4) | 85 (93.4) | 70 (94.6) |
| Asian | 272 (4.6) | 30 (4.7) | 4 (4.4) | 3 (4.1) |
| Black | 94 (1.6) | 5 (0.8) | 0 (0.0) | 0 (0.0) |
| Mixed Race | 70 (1.2) | 6 (0.9) | 1 (1.1) | 1 (1.4) |
| Other Ethnic Group | 52 (0.9) | 4 (0.6) | 1 (1.1) | 0 (0.0) |
| Prefer not to say | 16 (0.3) | 4 (0.6) | 0 (0.0) | 0 (0.0) |
| Medical condition | | | | |
| Chronic respiratory conditions | 738 (12.6) | 75 (11.7) | 13 (14.3) | 8 (10.8) |
| Chronic non-respiratory conditions | 679 (11.6) | 76 (11.8) | 8 (8.8) | 13 (17.6) |
| Immunosuppression | 128 (2.2) | 16 (2.5) | 2 (2.2) | 1 (1.4) |
| No medical condition | 4,318 (73.6) | 476 (74.0) | 68 (74.7) | 52 (70.3) |
| Household | | | | |
| Lives alone | 683 (11.6) | 82 (12.8) | 11 (12.1) | 9 (12.2) |
| Lives with adults | 3,010 (51.3) | 325 (50.5) | 42 (46.2) | 30 (40.5) |
| Lives with children | 2,170 (37.0) | 236 (36.7) | 38 (41.8) | 35 (47.3) |
| Staff type | | | | |
| Nursing | 1,957 (33.4) | 221 (34.4) | 22 (24.2) | 21 (28.4) |
| Administrative/Executive | 1,014 (17.3) | 95 (14.8) | 14 (15.4) | 11 (14.9) |
| Doctor | 670 (11.4) | 67 (10.4) | 12 (13.2) | 14 (18.9) |
| Healthcare Assistant | 360 (6.1) | 53 (8.2) | 3 (3.3) | 2 (2.7) |
| Healthcare Scientist | 279 (4.8) | 32 (5.0) | 7 (7.7) | 5 (6.8) |
| Student | 175 (3.0) | 25 (3.9) | 3 (3.3) | 1 (1.4) |
| Physiotherapist/Occupational Therapist/SALT | 228 (3.9) | 33 (5.1) | 10 (11.0) | 4 (5.4) |
| Midwife | 133 (2.3) | 12 (1.9) | 3 (3.3) | 1 (1.4) |
| Pharmacist | 129 (2.2) | 16 (2.5) | 3 (3.3) | 2 (2.7) |
| Estates/Porters/Security | 113 (1.9) | 8 (1.2) | 3 (3.3) | 1 (1.4) |
| Other | 805 (13.7) | 81 (12.6) | 11 (12.1) | 12 (16.2) |
| Occupation setting | | | | |
| Office | 1,333 (22.7) | 135 (21.0) | 26 (28.6) | 11 (14.9) |
| Outpatient | 1,213 (20.7) | 135 (21.0) | 19 (20.9) | 18 (24.3) |
| Inpatient Wards | 671 (11.4) | 89 (13.8) | 9 (9.9) | 13 (17.6) |

*(Continued)*

**Table 1.** (Continued)

| Characteristic | All participants n (%) | SARS-CoV-2 positive n (%) | Influenza positive n (%) | RSV positive n (%) |
|---|---|---|---|---|
| Patient facing (non-clinical) | 273 (4.7) | 29 (4.5) | 2 (2.2) | 8 (10.8) |
| Intensive Care | 208 (3.5) | 30 (4.7) | 4 (4.4) | 4 (5.4) |
| Theatres | 152 (2.6) | 22 (3.4) | 2 (2.2) | 5 (6.8) |
| Ambulance/Emergency Department | 108 (1.8) | 12 (1.9) | 4 (4.4) | 2 (2.7) |
| Maternity/Labour Ward | 82 (1.4) | 7 (1.1) | 1 (1.1) | 1 (1.4) |
| Other | 1,823 (31.1) | 184 (28.6) | 24 (26.4) | 12 (16.2) |
| Patient contact | | | | |
| Yes | 4,841 (82.6) | 535 (83.2) | 75 (82.4) | 71 (95.9) |
| No | 1,022 (17.4) | 108 (16.8) | 16 (17.6) | 3 (4.1) |
| Vaccination status | | | | |
| Both | 4,349 (74.2) | 481 (74.8) | 62 (68.1) | 54 (73.0) |
| Influenza 2022/23 vaccine only | 499 (8.5) | 53 (8.2) | 8 (8.8) | 5 (6.8) |
| COVID-19 second booster dose only | 354 (6.0) | 28 (4.4) | 10 (11.0) | 5 (6.8) |
| Neither | 661 (11.3) | 81 (12.6) | 11 (12.1) | 10 (13.5) |

Note: Eight participants had multiple infections during the analysis period – four participants had a SARS-CoV-2 and an influenza infection, three had influenza and RSV, and one had SARS-CoV-2 and RSV.

A higher proportion of participants with an RSV infection lived with children (47.3%) compared to those with a SARS-CoV-2 (36.7%) or an influenza (41.8%) infection. A higher proportion of participants with SARS-CoV-2 lived with adults (50.5%) compared to those with an influenza (46.2%) or an RSV (40.5%) infection (Table 1).

### SARS-CoV-2, influenza and RSV infection rates over winter 2022/23

There were 16 samples that were positive for more than one virus and were excluded from subsequent analyses: five samples were positive for SARS-CoV-2 and influenza, two samples were positive for SARS-CoV-2 and RSV, one sample was positive for influenza and RSV, and eight samples were positive for all three viruses.

Of the 26,476 samples collected, 98.8% (26,163) samples were testing for SARS-CoV-2, 73.4% (19,442) for influenza and 73.4% (19,445) for RSV.

There were 808 positive samples among 800 participants (643 SARS-CoV-2; 91 influenza; 74 RSV infections).

There were eight participants who were infected with more than one virus during the analysis period. Of these, four participants were infected with SARS-CoV-2 and influenza, three with influenza and RSV and one with SARS-CoV-2 and RSV. The median time between infections was 31 days (IQR: 18–61.5 days).

PCR positivity rates for all three viruses peaked in early December 2022 (peak positive PCR per 100 tests: 4.6 (95% CI 3.5, 5.7) for SARS-CoV-2, 3.9 (95% CI 2.2, 5.6) for influenza, 1.4 (95% CI 0.4, 2.4) for RSV), with influenza and RSV decreasing to low levels after January 2023 (for influenza ≤0.2 and RSV ≤ 0.3 positives per 100 tests), whereas SARS-CoV-2 maintained rates around 2.5 per 100 tests until March 2023 (Fig 2).

### Influenza-like illness symptoms over winter 2022/23

There were 4,003 (68.3%) participants who reported symptoms over the analysis period, with 1,054 (18.0%) reporting ILI symptoms. The proportion of participants reporting ILI symptoms by fortnight peaked in early December 2022 (peak fortnight was 7.5% (95% CI 6.8, 8.3) (368/4,893 surveys) (Fig 3).

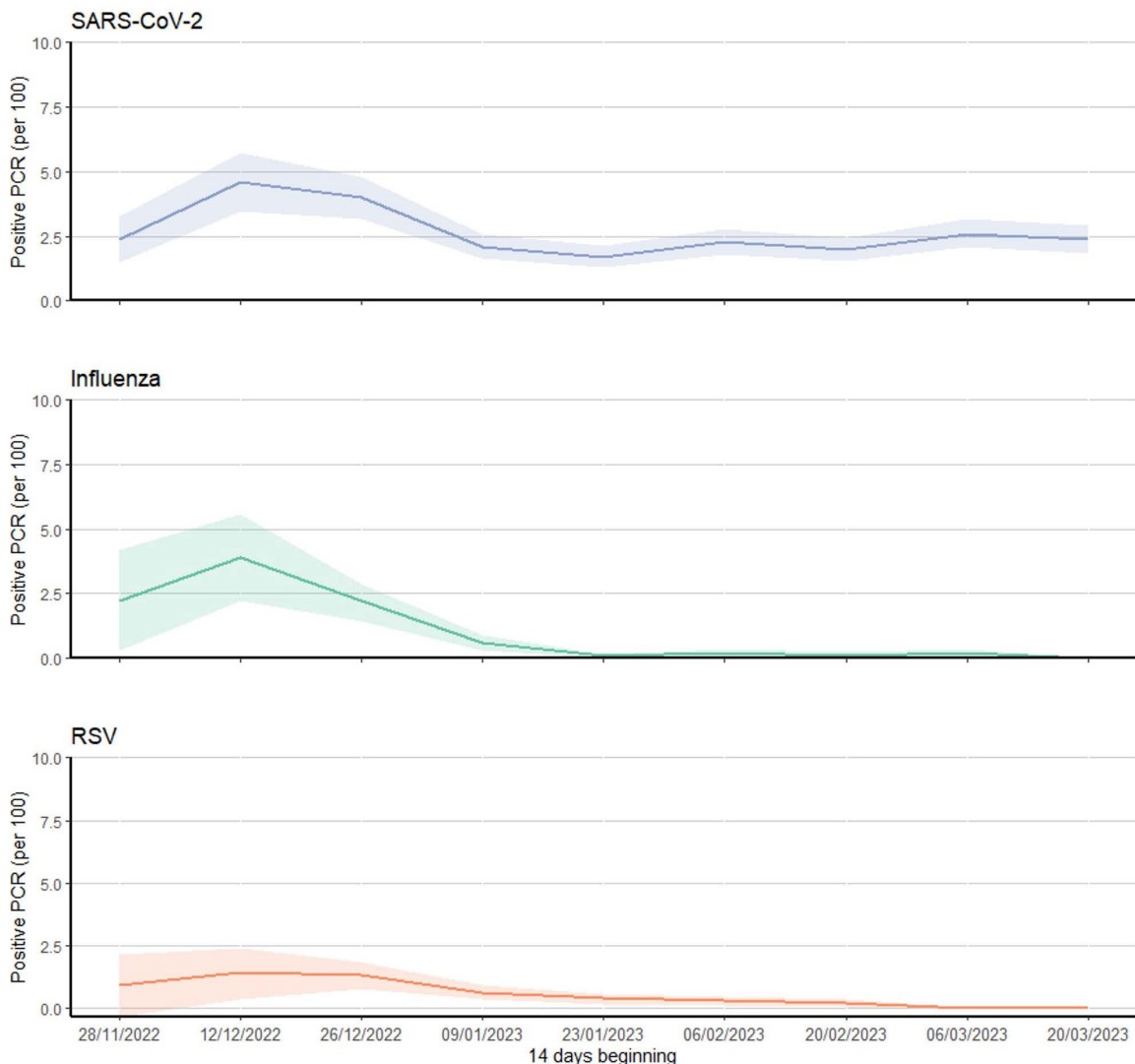

**Fig 2. PCR positivity rate (per 100 tests) by fortnight and infection, 28 November 2022 to 31 March 2023.** *Note: shaded areas represent 95% confidence intervals.*

## Symptom profile by SARS-CoV-2, influenza and RSV

The proportion of asymptomatic infections were similar among the three viruses – influenza (45.6%), SARS-CoV-2 (41.7%) and RSV (37.5%). Symptom profiles were similar for all three viruses. Although, loss of sense of smell and taste were more common among those with an SARS-CoV-2 infection and fever more common with a influenza infection. Hospital attendance was low for all viruses (Table 2).

## Time off work due to being symptomatic over winter 2022/23

Of participants testing positive for SARS-CoV-2 and RSV, ≥ 50% of participants reported taking time off work for being symptomatic (52.2% and 50.0%, respectively), compared to 28.6% for influenza infections (Table 2). The median number of days taken off work for a SARS-CoV-2 infection was 5 (IQR: 2–7); 3 days (IQR: 2.25–5) for influenza and 4 days (IQR: 2–7) for RSV.

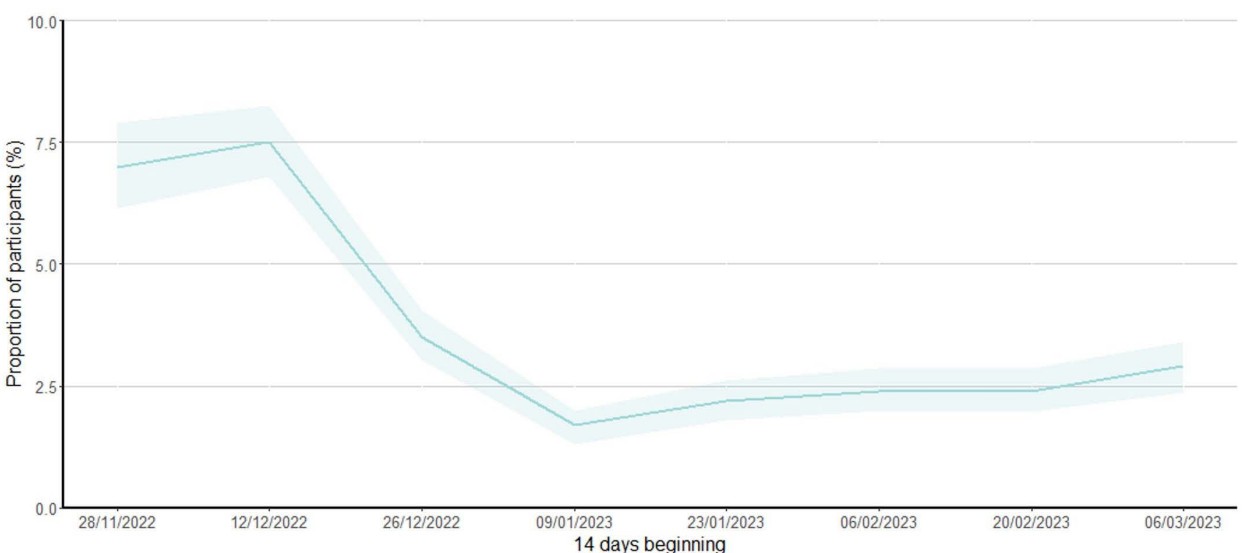

**Fig 3. Proportion of participants reporting influenza-like illness symptoms by fortnight, 28 November 2022 to 31 March 2023.** *Shaded areas represent 95% confidence intervals.*

**Table 2. Description of the symptom profile and severity of SARS-CoV-2, influenza and RSV PCR positive infections, 28 November 2022 to 31 March 2023.**

|  | SARS-CoV-2 n (%) | Influenza n (%) | RSV n (%) | *p-value (SARS-CoV-2 vs influenza)* | *p-value (SARS-CoV-2 vs RSV)* | *p-value (RSV vs influenza)* |
|---|---|---|---|---|---|---|
| **Total** | **643** | **93** | **74** |  |  |  |
| **Symptoms** |  |  |  | *0.494* | *0.594* | *0.327* |
| Any symptoms | 346 (58.3) | 49 (54.4) | 40 (62.5) |  |  |  |
| Asymptomatic | 247 (41.7) | 41 (45.6) | 24 (37.5) |  |  |  |
| **Symptoms type** |  |  |  |  |  |  |
| Loss of smell | 67 (11.3) | 3 (3.3) | 4 (6.3) | *0.015* | *0.289* | *0.450* |
| Cough | 182 (30.7) | 33 (36.7) | 17 (26.6) | *0.274* | *0.568* | *0.223* |
| Diarrhoea | 37 (6.2) | 5 (5.6) | 2 (3.1) | *>0.999* | *0.414* | *0.700* |
| Loss of taste | 80 (13.5) | 3 (3.3) | 4 (6.3) | *0.005* | *0.116* | *0.450* |
| Fatigue | 160 (27.0) | 24 (26.7) | 13 (20.3) | *>0.999* | *0.297* | *0.445* |
| Fever | 105 (17.7) | 25 (27.8) | 9 (14.1) | *0.030* | *0.602* | *0.050* |
| Swollen glands | 45 (7.6) | 6 (6.7) | 1 (1.6) | *>0.999* | *0.074* | *0.240* |
| Headache | 228 (38.4) | 29 (32.2) | 24 (37.5) | *0.294* | *>0.999* | *0.606* |
| Muscle aches | 179 (30.2) | 24 (26.7) | 13 (20.3) | *0.538* | *0.112* | *0.445* |
| Runny nose | 258 (43.5) | 33 (36.7) | 28 (43.8) | *0.253* | *>0.999* | *0.406* |
| Shortness of breath | 87 (14.7) | 12 (13.3) | 13 (20.3) | *0.873* | *0.270* | *0.273* |
| Sore throat | 231 (39.0) | 39 (43.3) | 29 (45.3) | *0.488* | *0.348* | *0.870* |
| Vomiting | 38 (6.4) | 8 (8.9) | 5 (7.8) | *0.368* | *0.598* | *>0.999* |
| **Duration of symptoms** |  |  |  | *0.090* | *0.405* | *0.729* |
| 1-3 | 43 (12.6) | 1 (2.0) | 2 (5.0) |  |  |  |
| 4-6 | 79 (23.2) | 10 (20.4) | 8 (20.0) |  |  |  |

*(Continued)*

**Table 2.** (Continued)

| | SARS-CoV-2 n (%) | Influenza n (%) | RSV n (%) | p-value (SARS-CoV-2 vs influenza) | p-value (SARS-CoV-2 vs RSV) | p-value (RSV vs influenza) |
|---|---|---|---|---|---|---|
| 7-14 | 54 (15.9) | 8 (16.3) | 9 (22.5) | | | |
| >14 | 164 (48.2) | 30 (61.2) | 21 (52.5) | | | |
| **Sick leave** | | | | *0.002* | *0.868* | *0.049* |
| Taken | 180 (52.2) | 14 (28.6) | 20 (50.0) | | | |
| Not taken | 165 (47.8) | 35 (71.4) | 20 (50.0) | | | |
| **Number of sick leave taken (days)** | | | | *0.015* | *0.484* | *0.079* |
| 1-3 | 67 (37.2) | 9 (64.3) | 10 (50.0) | | | |
| 4-6 | 56 (31.1) | 5 (35.7) | 4 (20.0) | | | |
| 7-14 | 57 (31.7) | 0 (0.0) | 6 (30.0) | | | |
| **Hospital attendance** | | | | *0.358* | *0.095* | *0.654* |
| Yes | 8 (2.3) | 2 (4.1) | 3 (7.5) | | | |
| No | 338 (97.7) | 47 (95.9) | 37 (92.5) | | | |

Over the analysis period, a total of 1,368/5,863 (23.3%) participants reported taking time off work due to being symptomatic; taking a median of 4 days off over the analysis period (range: 1–59 days). Of participants reporting ILI symptoms, 616/1,054 (58.4%) reported taking time off work; taking a median of 5 days; range: 1–59 days.

The 5,863 participants included contributed to a total of 507,388 follow-up days during the winter period. The rate of sick leave over this period was 1.48 days per 100 days of follow-up, with the peak seen in early December 2022 (2.4 days per 100 days) (Fig 4).

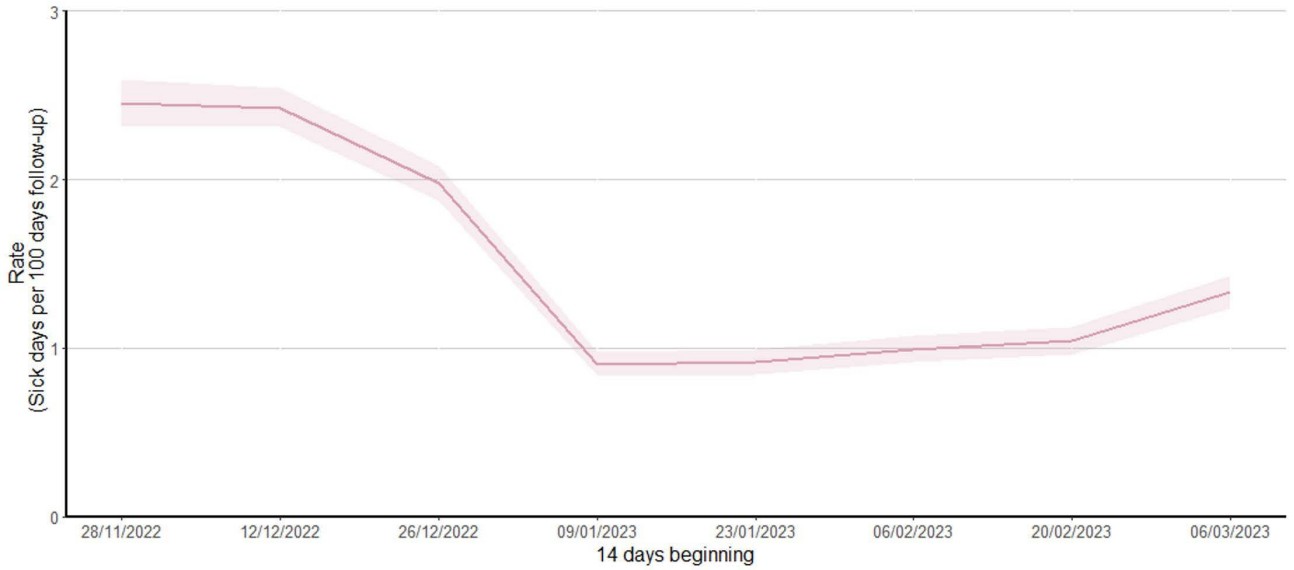

**Fig 4. Rate of sick leave taken (per 100 days) by fortnight, 28 November 2022 to 31 March 2023.** *Note: Sick leave rate was calculated as number of sick leave days taken divided by the number of days participants contributed to the analysis period, per 100 days; Shaded areas represent 95% confidence intervals.*

## Factors associated with taking time off work

Participants who took time off work due to being symptomatic between November 2022 and March 2023 appeared to differ by occupational setting, long-term medical conditions, and vaccination status (Table 3).

Participants in office-based roles took less time off work (1.11 days per 100 days) when compared to clinical roles such as theatres and intensive care units (2.13 and 1.91 days per 100 days, respectively). Estimated from adjusted Incidence Rate Ratios (IRR) also show this; theatres (1.89; 95% CI 1.66, 2.15) and intensive care units (1.74; 95% CI 1.54, 1.96) verses office-based participants. Participants with immunosuppressive, chronic respiratory and chronic non-respiratory conditions took more time off work than participants with no reported comorbidities (who took off 1.29 days per 100 days): immunosuppressive conditions 2.71 days per 100, adjusted IRR 2.19 (95% CI 1.95, 12.46); chronic respiratory 2.01/100 days, adjusted IRR 1.60 (95% CI 1.50, 1.70); and chronic non-respiratory conditions 1.87/100 days, adjusted IRR 1.52 (95% CI 1.42, 1.63).

**Table 3. Factors associated with time off work due to being symptomatic, 28 November 2022 to 31 March 2023.**

| Characteristic | Sick leave rate [b] (95% CI) | IRR (95% CI) | Adjusted IRR (95% CI) |
|---|---|---|---|
| **Vaccination status [a]** | | | |
| Both COVID-19 second booster and Influenza 2022/23 vaccine | 1.41 (1.37, 1.45) | Ref | Ref |
| Influenza 2022/23 Vaccine | 1.52 (1.40, 1.65) | 1.08 (0.99, 1.17) | 1.04 (0.95, 1.13) |
| COVID-19 second booster dose | 1.26 (1.14, 1.40) | 0.90 (0.81, 1.00) | 0.91 (0.82, 1.02) |
| Neither vaccine | 2.04 (1.93, 2.16) | 1.45 (1.36, 1.54) | 1.47 (1.38, 1.56) |
| **Age group** | | | |
| Under 25 | 1.55 (1.10, 2.16) | Ref | Ref |
| 25–34 | 1.84 (1.69, 1.99) | 1.19 (0.85, 1.66) | 1.16 (0.83, 1.63) |
| 35–44 | 1.51 (1.44, 1.59) | 0.98 (0.70, 1.36) | 0.92 (0.66, 1.28) |
| 45–54 | 1.49 (1.43, 1.54) | 0.96 (0.69, 1.33) | 0.91 (0.65, 1.26) |
| 55–64 | 1.44 (1.38, 1.50) | 0.93 (0.67, 1.29) | 0.89 (0.64, 1.24) |
| Over 65 | 0.69 (0.56, 0.84) | 0.44 (0.30, 0.65) | 0.45 (0.30, 0.66) |
| **Occupation setting** | | | |
| Theatres | 2.13 (1.89, 2.40) | 1.91 (1.68, 2.18) | 1.89 (1.66, 2.15) |
| Intensive Care | 1.91 (1.72, 2.13) | 1.72 (1.52, 1.94) | 1.74 (1.54, 1.96) |
| Outpatient | 1.62 (1.55, 1.70) | 1.45 (1.35, 1.56) | 1.49 (1.39, 1.60) |
| Ambulance/Emergency Department | 1.58 (1.33, 1.86) | 1.42 (1.19, 1.69) | 1.36 (1.14, 1.62) |
| Maternity/Labour Ward | 1.58 (1.30, 1.92) | 1.42 (1.16, 1.73) | 1.38 (1.13, 1.68) |
| Inpatient Wards | 1.57 (1.47, 1.67) | 1.41 (1.29, 1.53) | 1.36 (1.25, 1.49) |
| Patient facing (non-clinical) | 1.53 (1.38, 1.70) | 1.38 (1.23, 1.55) | 1.40 (1.25, 1.57) |
| Other | 1.50 (1.44, 1.56) | 1.35 (1.26, 1.44) | 1.38 (1.29, 1.47) |
| Office | 1.11 (1.05, 1.18) | Ref | Ref |
| **Medical group** | | | |
| No medical condition | 1.29 (1.26, 1.33) | Ref | Ref |
| Immunosuppression | 2.71 (2.42, 3.03) | 2.09 (1.86, 2.35) | 2.19 (1.95, 2.46) |
| Chronic Respiratory conditions | 2.01 (1.90, 2.12) | 1.55 (1.46, 1.65) | 1.60 (1.50, 1.70) |
| Chronic Non-Respiratory conditions | 1.87 (1.77, 1.99) | 1.45 (1.36, 1.55) | 1.52 (1.42, 1.63) |

[a] Participants were excluded if they were vaccinated in one survey period and unvaccinated in another survey period. Surveys were excluded from the analysis if the participant was vaccinated within the survey date range. [b] Sick leave rate was calculated as number of sick leave days taken divided by the number of days participants contributed to the analysis period, per 100 days. IRR = Incidence rate ratio. CI = Confidence interval.

Those who were vaccinated against both influenza and COVID-19 took less time off work (1.41 days per 100 days) than those who did not receive either vaccine (2.04 days per 100 days) (Table 3). After controlling for age, occupational setting and co-morbidities, we estimated that those who received neither vaccine had a 47% higher rate of sick leave than those vaccinated for both (adjusted IRR 1.47 (95% CI 1.38, 1.56)).

## Discussion

Results from the 2022/23 SIREN Winter Pressures pilot study demonstrate the impact of respiratory illness on the NHS workforce. Rates of SARS-CoV-2, influenza and RSV, and symptoms, all peaked in early December, resulting in increased levels of sickness absence over this time period.

The PCR positivity trends over winter 2022/23 in our cohort of HCW is consistent with national surveillance data [21] demonstrating the potential utility of conducting surveillance in this population to determine the prevalence and impact of respiratory viral infections in the working age population [24]. Our detection of RSV infections in this population, which is rarely tested for this virus, suggests that RSV also contributes to winter pressures. The finding that symptom profiles were similar across the three viruses is consistent with existing published literature [25]. Across all three viruses, a substantial proportion of infections were asymptomatic but could potentially contribute to transmission, in particular, influenza with over two-thirds of infections asymptomatic [26]. This highlights the potential risk associated with nosocomial infections in healthcare workers, though we do not have strong evidence how transmissible asymptomatic infections is. The impact of respiratory illness on time off work in our cohort of HCW over winter 2022/23 was considerable, with 23% of participants reporting time off work due to being symptomatic, and all three viruses contributing to this. Sick leave rates varied by demographics and vaccination status, with lower estimated rates among those who received the second COVID-19 booster and seasonal influenza vaccine. Given the impact that HCW time off work could have on healthcare resilience over winter, further research into associated factors, including behaviours and attitudes, is important.

A key limitation of this pilot study was the coverage and timing of multiplex PCR roll-out. There were fewer participants with PCR results for influenza and RSV than those with a SARS-CoV-2 test result for two main reasons. Firstly, due to technical difficulties in reporting swab results through newly established laboratories systems, a small proportion of results for influenza and RSV were not able to be linked to the SARS-CoV-2 result for the individual swab. Secondly, as this was a pilot, multiplex testing was introduced late in the winter season, with testing data only available from late November 2022, due to delays switching PCR platforms across NHS laboratories (using a decentralised study testing model), and the timing of establishing a new centralised postal PCR pathway. This delay was compounded by an unusually early influenza season in 2022/23 [21] Consequently, our surveillance of these viruses over winter 2022/23 may have missed some infections within our cohort, particularly influenza and RSV.

The SIREN cohort has a high proportion of female participants, those of white ethnicity and a median age over 45 years, this is broadly similar to the NHS workforce [27–29] although this analysis only consists a small proportion of the whole workforce.

This pilot sub-study conducted during winter 2022/23 has demonstrated the adaptability and applicability of findings from the SIREN HCW cohort. Results from the study show the benefit of regular multiplex testing across NHS Trusts to understand the interplay and impact of seasonal viruses on workforce planning, patient care and healthcare resilience in the NHS during the Winter. We have demonstrated that all three viruses contribute to staff illness and time off work over winter, and that seasonal flu and COVID-19 vaccines were associated with lower sick leave rates. Future studies should consider using a centralised multiplex testing pathway to improve surveillance of respiratory infections and ensure the timing of testing is optimised.

## Acknowledgments

Our thanks go to the participants in the SIREN study as well as the site research teams. We would also like to thank the help and support offered by the Berkshire Research Ethics Committee. As well as the members of the SIREN study group.

Lead author for the SIREN study group: Victoria Hall. Email: Victoria.Hall@ukhsa.gov.uk

SIREN study group: John Northfield (Site research team), Sean Cutler (Site research team), Anna Roynon (Site research team), Maxine Nash (Site research team), Amanda Dell (Site research team), Louise Parfitt (Site research team), Andrea Richards (Site research team), Andrea Price (Site research team), Christian Subbe (Site research team), Caroline Mulvaney Jones (Site research team), Julia Roberts (Site research team), Manny Bagary (Site research team), Nadezda Starkova (Site research team), Inderpreet Athwal (Site research team), Louise Hudson (Site research team), Ashley Jones (Site research team), Rebecca Chapman (Site research team), Lucy Booth (Site research team), Claire Williams (Site research team), Fiona Adair (Site research team), April Hawkins (Site research team), Chinari Subudhi (Site research team), Scott Latham (Site research team), Raksha Mistry (Site research team), Natalie Silva (Site research team), Abigail Severn (Site research team), Alejandro Arenas-Pinto (Site research team), Eva McAlpine (Site research team), Aran Dhillon (Site research team), Connor McAlpine (Site research team), Gosala Gopalakrishnan (Site research team), Sarah Creer (Site research team), Eve Etell Kirby (Site research team), Kim Gray (Site research team), Joanna Wright (Site research team), Joely Morgan (Site research team), Gemma Harrison (Site research team), Mark Broadhurst (Site research team), Simon Taylor (Site research team), Clare McAdam (Site research team), Natalie Crooks (Site research team), Stacey Horne (Site research team), Anna Grice (Site research team), Nicola Walker (Site research team), Luke Bedford (Site research team), Paul Ridley (Site research team), Alison O'Kelly (Site research team), Catherine Sinclair (Site research team), Val Irvine (Site research team), Elizabeth Boyd (Site research team), Claire Thomas (Site research team), Ina Hoad (Site research team), Tryphena Konala (Site research team), Judith Radmore (Site research team), Emily Macnaughton (Site research team), Sarah Knight (Site research team), Kim Hulacka (Site research team), Robert Shorten (Site research team), Kathryn Hollinshead (Site research team), Lois Bullen (Site research team), Robert Shorten (Site research team), Claire Corless (Site research team), Sarah Mcloughlin (Site research team), Bethany Preece (Site research team), Sarah Baillon (Site research team), Samantha Hamer (Site research team), Joanne Edgar (Site research team), Kelly Moran (Site research team), Vijayendra Waykar (Site research team), Charlotte Wesson (Site research team), Rebecca Rutter (Site research team), Maureen Williams (Site research team), Bethany Jones (Site research team), Russell Coram (Site research team), Holly Slater (Site research team), Joanne Jones (Site research team), Banher Sandhu (Site research team), Elijah Matovu (Site research team), Claire Gabriel (Site research team), Katherine Pagett (Site research team), Sheron Clarke (Site research team), Sally Mavin (Site research team), Sebastien Fagegaltier (Site research team), Shannon Proctor (Site research team), Mary Summerscales (Site research team), Andrew Gibson (Site research team), Alexandra Cochrane (Site research team), Dawid Dytmer (Site research team), Lita Kovina (Site research team), Grace Davies (Site research team), Manish Patel (Site research team), Berni Welsh (Site research team), Karen Black (Site research team), Kate Templeton (Site research team), Sam Donaldson (Site research team), Andrea Clarke (Site research team), Jane Crowe (Site research team), Kadiga Campbell (Site research team), Barbara Hamilton (Site research team), Liz Sheridan (Site research team), Charlotte Barclay (Site research team), Maxine Ashton (Site research team), Alison Rodger (Site research team), Tabitha Mahungu (Site research team), Debbie Delgado (Site research team), Julia Vasant (Site research team), Deborah Howcroft (Site research team), Sarah Meisner (Site research team), Abby Rand (Site research team), Catherine Thompson (Site research team), Sophia Strong-Sheldrake (Site research team), Vicky King (Site research team), Emma Underhill (Site research team), Kate Seymour (Site research team), Holly Morgan (Site research team), Ash Turner (Site research team), Anne Hayes (Site research team), Masood Aga (Site research team), James Pethick (Site research team), Ashok Dadrah (Site research team), Thushan de Silva (Site research team), Helen Shulver (Site research team), Gareth Stephens (Site research team), Simon Tazzyman (Site research team), Mandy Carnahan (Site research team), Mandy Beekes (Site research team), Sanal Jose (Site research team), Jo stickley (Site research team), Hannah Gibson (Site research team), Yuri Protaschik (Site research team), Susan Regan (Site research team), Alison Campbell (Site research team), John Day (Site research team), Swapna Kunhunny (Site research team), Bernard Hadebe (Site research team), Paula Harman (Site research team), Sharon Tysoe (Site research team),

Bridgett Masunda (Site research team), Nigara Atayeva (Site research team), Joanne Galliford (Site research team), Prisca Gondo (Site research team), Raji Orath Prabakaran (Site research team), Jane Dare (Site research team), Qi Zheng (Site research team), Danielle McCracken (Site research team), Emmanuel Defever (Site research team), Ellene Thompson (Site research team), Lynda Fothergill (Site research team), Karen Burns (Site research team), Andrew Higham (Site research team), Lisa Bishop (Site research team), Aileen Menzies (Site research team), Matt Horton (Site research team), Therese Kelly (Site research team), Cristina Dragu (Site research team), David Hilton (Site research team), Hannah Jory (Site research team), Penny Harris (Site research team), Susan Hopkins (UKHSA SIREN team), Victoria Hall (UKHSA SIREN team), Jasmin Islam (UKHSA SIREN team), Ana Atti (UKHSA SIREN team), Omoyeni Adebiyi (UKHSA SIREN team), Nick Andrews (UKHSA SIREN team), Hannah Emmett (UKHSA SIREN team), Jonathan Broad (UKHSA SIREN team), Nish Kapirial (UKHSA SIREN team), Simone Dyer (UKHSA SIREN team), Sophie Russell (UKHSA SIREN team), Colin Brown (UKHSA SIREN team), Joanna Conneely (UKHSA SIREN team), Paul Conneely (UKHSA SIREN team), Sarah Foulkes (UKHSA SIREN team), Nabila Fowles-Gutierrez (UKHSA SIREN team), Nipunadi Hettiarachchi (UKHSA SIREN team), Jameel Khawam (UKHSA SIREN team), Edward Monk (UKHSA SIREN team), Katie Munro (UKHSA SIREN team), Andrew Taylor-Kerr (UKHSA SIREN team), Jean Timeyin (UKHSA SIREN team), Edgar Wellington (UKHSA SIREN team), Angela Dunne (UKHSA SIREN team), Dominic Sparkes (UKHSA SIREN team), Naomi Platt (UKHSA SIREN team), Anna Howells (UKHSA SIREN team), Enemona Adaji (UKHSA SIREN team), Omolola Akinbami (UKHSA SIREN team), Palak Joshi (UKHSA SIREN team), Paola Barbero (UKHSA SIREN team), Meera Chand (UKHSA SIREN team), Andre Charlett (UKHSA SIREN team), Michelle Cole (UKHSA SIREN team), Claire Neill (UKHSA SIREN team), Anne-Marie O'Connell (UKHSA SIREN team), Ferdinando Insalata (UKHSA SIREN team), Tim Brooks (UKHSA SIREN team), Maria Zambon (UKHSA SIREN team), Mary Ramsay (UKHSA SIREN team), Ayoub Saei (UKHSA SIREN team), Ezra Linley (UKHSA SIREN team), Simon Tonge (UKHSA SIREN team), Ashley Otter (UKHSA SIREN team), Silvia D'Arcangelo (UKHSA SIREN team), Cathy Rowe (UKHSA SIREN team), Amanda Semper (UKHSA SIREN team), Eileen Gallagher (UKHSA SIREN team), Robert Kyffin (UKHSA SIREN team), Kate Howell (UKHSA SIREN team), Jacqueline Hewson (UKHSA SIREN team), Iain Milligan (UKHSA SIREN team), Noshin Sajedi (UKHSA SIREN team), Davina Calbraith (UKHSA SIREN team), Caio Tranquillini (UKHSA SIREN team), Jerry Ye Aung Kyaw (UKHSA SIREN team), Lisa Cromey (Public Health Agency Northern Ireland), Dianne Corrigan (Public Health Agency Northern Ireland), Desmond Areghan (Glasgow Caledonian University), Jennifer Bishop (Public Health Scotland), Melanie Dembinsky (Glasgow Caledonian University), Laura Dobbie (Public Health Scotland), Josie Evans (Public Health Scotland), David Goldberg (Public Health Scotland), Lynne Haahr (Glasgow Caledonian University & Public Health Scotland), Annelysse Jorgenson (Glasgow Caledonian University), Ayodeji Matuluko (Glasgow Caledonian University), Laura Naismith (Public Health Scotland), Desy Nuryunarsih (Glasgow Caledonian University & Public Health Scotland), Alexander Olaoye (Glasgow Caledonian University), Caitlin Plank (Public Health Scotland), Lesley Price (Glasgow Caledonian University & Public Health Scotland), Nicole Sergenson (Glasgow Caledonian University & Public Health Scotland), Sally Stewart (Glasgow Caledonian University & Public Health Scotland), Andrew Telfer (Public Health Scotland), Jennifer Weir (Public Health Scotland), Ellen De Lacy (Public Health Wales), Yvette Ellis (Health and Care Research Wales), Susannah Froude (Public Health Wales), Chris Norman (Health and Care Research Wales), Guy Stevens (Public Health Wales), Linda Tyson (Public Health Wales).

## Author contributions

**Conceptualization:** Sarah Foulkes, Conall H Watson, Susan Hopkins, Victoria Hall.

**Data curation:** Sarah Foulkes, Katie Munro, Jameel Khawam.

**Formal analysis:** Sarah Foulkes, Katie Munro.

**Funding acquisition:** Victoria Hall.

**Investigation:** Sarah Foulkes.

**Methodology:** Sarah Foulkes, Nick Andrews, Andre Charlett, Victoria Hall.

**Project administration:** Dominic Sparkes, Jonathan Broad, Naomi Platt, Anna Howells, Omolola Akinbami, Sophie Russell, Chris Norman, Lesley Price, Diane Corrigan, Michelle Cole, Jean Timeyin, Louise Forster, Ana Atti, Jasmin Islam, Colin S Brown, Susan Hopkins, Victoria Hall.

**Resources:** Jameel Khawam, Palak Joshi, Katrina Slater, Jonathan Turner.

**Supervision:** Sarah Foulkes, Susan Hopkins, Victoria Hall.

**Validation:** Sarah Foulkes, Katie Munro, Jameel Khawam.

**Visualization:** Sarah Foulkes, Katie Munro.

**Writing – original draft:** Sarah Foulkes, Katie Munro, Dominic Sparkes, Victoria Hall.

**Writing – review & editing:** Sarah Foulkes, Katie Munro, Dominic Sparkes, Jonathan Broad, Naomi Platt, Anna Howells, Omolola Akinbami, Jameel Khawam, Palak Joshi, Sophie Russell, Chris Norman, Lesley Price, Diane Corrigan, Michelle Cole, Jean Timeyin, Louise Forster, Katrina Slater, Conall H Watson, Nick Andrews, Andre Charlett, Ana Atti, Jasmin Islam, Colin S Brown, Jonathan Turner, Susan Hopkins, Victoria Hall.

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
