## [Decision Letter · Decision Letter 0]

11 Dec 2024

PONE-D-24-39131Adapting COVID-19 research infrastructure to capture influenza and respiratory syncytial virus alongside SARS-CoV-2 in UK healthcare workers winter 2022/23: Results of a pilot study in the SIREN cohortPLOS ONE

Dear Dr. Foulkes,

Thank you for submitting your manuscript to PLOS ONE. After careful consideration, we feel that it has merit but does not fully meet PLOS ONE’s publication criteria as it currently stands. Therefore, we invite you to submit a revised version of the manuscript that addresses the points raised during the review process.

**The Authors are expected to address all comments by the Reviewer. In particular, please provide the rationale of excluding samples with detection of multiple viruses, and describe the testing schedule in the Methods. In additional to the above comments, please address,**

**As a potential benefit of using multiplex testing, if sample size allows please consider the value of analyzing or providing some information on patients with co-detection of viruses.**

We look forward to receiving your revised manuscript.

Kind regards,

Eric HY Lau, Ph.D.

Academic Editor

PLOS ONE

**Journal Requirements:**

The SIREN study was funded by the UK Health Security Agency; the UK Department of Health and Social Care with contributions from the governments in Northern Ireland, Wales, and Scotland; the National Institute for Health Research; Health Data Research UK (NIHR200927; HDRUK2022.0322).

Our thanks go to the participants in the SIREN study. We would also like to thank the help and support offered by the Berkshire Research Ethics Committee and the HDR-UK for winter pressures funding and to UKHSA for core SIREN study funding. Thanks also to all research team and site staff not mentioned above.

The SIREN study was funded by the UK Health Security Agency; the UK Department of Health and Social Care with contributions from the governments in Northern Ireland, Wales, and Scotland; the National Institute for Health Research; Health Data Research UK (NIHR200927; HDRUK2022.0322).

4. In the online submission form, you indicated that Anonymised data will be made available for secondary analysis to trusted researchers upon reasonable request.

5. One of the noted authors is a group or consortium "SIREN Study Group". In addition to naming the author group, please list the individual authors and affiliations within this group in the acknowledgments section of your manuscript. Please also indicate clearly a lead author for this group along with a contact email address.

**Additional Editor Comments:**

The Authors are expected to address all comments by the Reviewer. In particular, please provide the rationale of excluding samples with detection of multiple viruses, and describe the testing schedule in the Methods. In additional to the above comments, please address,

1. As a potential benefit of using multiplex testing, if sample size allows please consider the value of analyzing or providing some information on patients with co-detection of viruses.

Reviewers' comments:

Reviewer's Responses to Questions

**Comments to the Author**

1. Is the manuscript technically sound, and do the data support the conclusions?

Reviewer #1: Partly

2. Has the statistical analysis been performed appropriately and rigorously? 

Reviewer #1: I Don't Know

3. Have the authors made all data underlying the findings in their manuscript fully available?

Reviewer #1: Yes

4. Is the manuscript presented in an intelligible fashion and written in standard English?

Reviewer #1: Yes

5. Review Comments to the Author

**Reviewer #1:**  I enjoyed reading the paper “Adapting COVID-19 research infrastructure to capture influenza and respiratory syncytial virus alongside SARS-CoV-2 in UK healthcare workers winter 2022/23: Results of a pilot study in the SIREN cohort”. Although the paper could use some more “focus” – I would suggest to use a couple of main topics and discuss them in every part of the paper in the same order. Consider to put more attention on RSV as that is new(er) information.

ABSTRACT

1.Please specify participants a bit further in abstract when possible “5,863 participants were included, 84.6% female, 70.3% ≥45-years, and 33.4% were nurses.”

INTRODUCTION

2.The introduction does not read “easily” in my opinion as the authors do not introduce the three viruses & there epidemiology separately at the beginning. As far as I am aware RSV has a different seasonal variation and also a different vaccination approach. And the effect(s) of influenza (vaccination) in HCW (including sick leave & transmission) is probably much more investigated than described by the authors and more than for RSV & SARS-CoV-2.

3. Consider describing the status of RSV vaccination protocols in the UK

METHODS

4.How many samples were positive for two or more viruses and why did the authors choose to exclude these? “samples which were positive for multiple viruses were excluded from the analysis”

5.Describe the testing schedule (see my comment later at the discussion session)

RESULTS

6.Main reasons for exclusion are missing

7.Please describe the study population in more detail especially the 70% who are not nurses in the results section

8.Was there any effect of household composition? Especially having young children?

DISCUSSION

9.Please mention this differences in testing schedules in the method section “A key limitation of this pilot study was the coverage and timing of multiplex PCR roll-out. There were fewer participants with PCR results for influenza and RSV than those with a SARS-Cov 2 test result”

10. I miss explanations or ideas concerning the study population which mostly consists of women and participants over 45 “Most participants were female (84.6%), of white ethnicity (91.4%), over 45-years of age (70.3%); nursing was the largest staff group (33.4%)”

6. PLOS authors have the option to publish the peer review history of their article (what does this mean? ). If published, this will include your full peer review and any attached files.

**Do you want your identity to be public for this peer review?** For information about this choice, including consent withdrawal, please see our Privacy Policy .

Reviewer #1: No

---

## [Author Response · Author response to Decision Letter 1]

7 Mar 2025

Responses to the reviewer and editor comments are included in the Response to Reviewers document, which has been uploaded to this re-submission.

---

## [Editor Report · Decision Letter 1]

13 Apr 2025

Adapting COVID-19 research infrastructure to capture influenza and respiratory syncytial virus alongside SARS-CoV-2 in UK healthcare workers winter 2022/23: Results of a pilot study in the SIREN cohort

PONE-D-24-39131R1

Dear Dr. Foulkes,

We’re pleased to inform you that your manuscript has been judged scientifically suitable for publication and will be formally accepted for publication once it meets all outstanding technical requirements.

Kind regards,

Eric HY Lau, Ph.D.

Academic Editor

PLOS ONE

Additional Editor Comments (optional):

Thanks for addressing all the editor’s and reviewers' comments. Congratulations on the excellent work!
---

## [Editor Report · Acceptance letter]

PONE-D-24-39131R1

PLOS ONE

Dear Dr. Foulkes,

I'm pleased to inform you that your manuscript has been deemed suitable for publication in PLOS ONE. Congratulations! Your manuscript is now being handed over to our production team.

Kind regards,

on behalf of

Dr. Eric HY Lau

Academic Editor

PLOS ONE